# Attribute Graphs Underlying Molecular Generative Models: Path to Learning with Limited Data

**Samuel C. Hoffman**  *shoffman@ibm.com*
*IBM Research*

**Payel Das**  *daspa@us.ibm.com*
*IBM Research*

**Karthikeyan Shanmugam**  *karthikeyanvs@google.com*
*IBM Research**

**Kahini Wadhawan**  *kahini.wadhawan1@ibm.com*
*IBM Research*

**Prasanna Sattigeri**  *psattig@us.ibm.com*
*IBM Research*

**Reviewed on OpenReview:** *https://openreview.net/forum?id=APON4bslQC*

## Abstract

Training generative models that capture rich semantics of the data and interpreting the latent representations encoded by such models are very important problems in un-/self-supervised learning. In this work, we provide a simple algorithm that relies on perturbation experiments on latent codes of a pre-trained generative autoencoder to uncover an attribute graph that is implied by the generative model. We perform perturbation experiments to check for influence of a given latent variable on a subset of attributes. Given this, we show that one can fit an effective graphical model that models a structural equation model between latent codes taken as exogenous variables and attributes taken as observed variables. One interesting aspect is that a single latent variable controls multiple overlapping subsets of attributes unlike conventional approaches that try to impose full independence. Using a pre-trained generative autoencoder trained on a large dataset of small molecules, we demonstrate that the graphical model between various molecular attributes and latent codes learned by our algorithm can be used to predict a specific property for molecules which are drawn from a different distribution. We compare prediction models trained on various feature subsets chosen by simple baselines, as well as existing causal discovery and sparse learning/feature selection methods, with the ones in the derived Markov blanket from our method. Results show empirically that the predictor that relies on our Markov blanket attributes is robust to distribution shifts when transferred or fine-tuned with a few samples from the new distribution, especially when training data is limited.

## 1 Introduction

While deep learning models demonstrate impressive performance in different prediction tasks, they often leverage correlations to learn a model of the data. As a result, accounting for the spurious ones among those correlations present in the data, which can correspond to biases, can weaken the robustness of a predictive model on unseen domains with distributional shifts. Such failure can lead to serious consequences when deploying machine learning models in the real world.

---

*currently at Google Research India

One way to obtain better generalization across domains is by learning causal relations in data, the core idea being to decipher the causal mechanism that is invariant across domains of interest. Structural causal models have provided a principled framework for inferring causality from the data alone which yields a better way to understand the data as well as the mechanisms underlying the generation of the data (Pearl, 2009; Shimizu et al., 2006). In this framework, causal relationships among variables are represented with a Directed Acyclic Graph (DAG).

While learning structured causal models from observational data alone is possible for tabular data (Chickering, 2020; Kalisch & Bühlman, 2007; Tsamardinos et al., 2006), in many applications involving complex modalities, such as images, text, and biological sequencing data, it may not be reasonable to directly learn a causal graph on the observed covariates. For example, in the case of image data, the observed pixels may hide the structure of the causal generative factors (Chalupka et al., 2014). In those cases, it may suffice to learn a structural model of the data to serve the purpose of domain generalization, which may not capture the true causal factors. Further, in datasets with complex modalities, limited knowledge about metadata (additional attributes) of the data samples may be readily available that can guide construction of such a structural model.

For real-world applications with complex modalities it is critical to learn a model of the high-dimensional dataset by using the limited labeled metadata available. This also helps in deriving a model of the data that provides better interpretability. We focus on extracting a graphical model between the latent codes of a generative model and attributes present in the metadata (even if limited) by leveraging a generative model trained on vast amounts of unsupervised data. Notably, since any accompanying metadata never exhaustively lists all underlying generating factors of a data point (e.g., image), our procedure allows for unobserved latents between attributes in the metadata.

In this work, we propose a simple alternative method to extract a graph structure relating different data attributes to latent representations learned by a given generative model trained on observational data. The generative model is pre-trained on large amount of unlabeled data. The domain-specific graph is then extracted using a smaller set of data with easily obtained labels. We then estimate the sensitivity of each of the data attributes by perturbing one latent dimension at a time and use that sensitivity information to construct a graphical model of attributes. The model's generalizability is finally tested by predicting an attribute on samples from a target distribution different from the source training data. Additionally, we are interested in scenarios where there is only limited data available from the target/test distribution. Therefore, we intend to learn the most data-efficient and accurate predictor, given a pre-trained generative model, to solve the downstream task.

To showcase the usefulness of the proposed framework, we focus on predicting a set of pharmacokinetic properties of organic small molecules. Specifically, the out-of-distribution (OOD) small molecules differ in term of chemical scaffolds (core components).

Below, we list our contributions:

1. We propose a novel method to learn the underlying graph structure between latent representations of a pre-trained generative model and attributes accompanying the dataset allowing for confounding between attributes.

2. We use the structure of the learnt model to do feature selection for predicting properties of out-of-distribution samples in a limited data setting.

3. We show that the graphical model learned by our method helps to achieve better generalization and is robust to distribution shifts.

4. The proposed method offers a means to derive an optimal representation from a pre-trained generative model, which enables sample-efficient domain adaptation, while providing (partial) interpretability.

## 2 Background and related work

### 2.1 Disentangled/Invariant generative models

Several prior works (Higgins et al., 2017; Makhzani et al., 2015; Kumar et al., 2017; Kim & Mnih, 2018) have considered learning generative models with a *disentangled* latent space. Many of them define disentanglement as the independence of the latent representation, i.e., changing one dimension/factor of the latent representation does not affect others. This involves additional constraints on the latent distribution during training as opposed to our method.

More generally, disentanglement requires that a primitive transformation in the observed data space (such as translation or rotation of images) results in a sparse change in the latent representation. In practice, the sparse changes can need not be *atomic*, i.e., a change in the observed space can lead to small but correlated latent factors. Additionally, it has been shown that it maybe impossible to achieve disentanglement without proper inductive biases or a form of supervision (Locatello et al., 2019). In Träuble et al. (2021), the authors use small number of ground truth latent factor labels and impose desired correlational structure to the latent space. Similarly, we extract a graphical model between latent representations of the generative model and the attributes that are available in the form of metadata.

Discovering the causal structure of the latent factors is an even more challenging but worthwhile pursuit compared to discovering only the correlated latent factors as it allows us to ask causal questions such as ones arising from interventions or counterfactuals. Causal GAN (Kocaoglu et al., 2017) assumes that the the causal graph of the generative factors is known a priori and learns the functional relations by learning a good generative model. CausalVAE (Yang et al., 2021) shows that under certain assumptions, a linear structural causal model (SCM) on the latent space can be identified while training the generative model. In contrast, here, we learn the graphical structure hidden in the pre-trained generative model in a post-hoc manner which does not affect the training of the generative model. Further, while the derived model shows domain generalization, indicating learning of invariant structure across distributions, it does not focus on capturing true causality. Therefore, our work is different from Besserve et al. (2019), as our goal is to learn a graphical model in the latent space instead of the causal structure hidden in the decoder of the generative model. Leeb et al. (2023) proposes a latent response framework, where the intervention in the latent variables reveals causal structure of the learned generative process as well as the relations between latent variables. The present work is distinct, as it aims to capture the response in the meta attribute space upon intervention on a latent dimension.

### 2.2 Invariant representation learning

Among conceptually related works, Arjovsky et al. (2019) focuses on learning invariant correlations across multiple training distributions to explore causal structures underlying the data and allow OOD generalization. To achieve this, they propose to find a representation $\phi$ such that the optimal classifier given $\phi$ is invariant across training environments. We leverage the fact that the pre-training encompasses data from a wide set of subdomains however our method does not require multiple explicit training environments. Dubois et al. (2020) propose a decodable information bottleneck objective to find an optimal set of predictors that are maximally informative of the label of interest, while containing minimal additional information about the dataset under consideration to avoid overfitting. Our work instead aims to derive an interpretable representation of the data using fixed attributes and guided by a learned graphical model.

### 2.3 Generative autoencoders for molecular representation learning

In recent years, generative autoencoders have emerged as a popular and successful approach for modeling both small and macromolecules (e.g., peptides) (Gómez-Bombarelli et al., 2018; Das et al., 2021; Chenthamarakshan et al., 2020; Hoffman et al., 2021). Often, those generative models are coupled with search or sampling methods to enforce design of molecules with certain attributes. Inspired by the advances in text generation (Hu et al., 2017) and the widely used text annotations of molecules, many of those frameworks imposed structure in the latent space by semi-supervised training or discriminator learning with metadata/labels. A

number of studies also have provided wet lab testing results of the machine-designed molecules derived from those deep generative foundation models, confirming the validity of the proposed designs (Nagarajan et al., 2017; Das et al., 2021; Shin et al., 2021; Chenthamarakshan et al., 2023). This demonstrates the effectiveness of generative models for molecular representations and implies that they have learned important insights about the domain that may be extracted.

## 2.4 Interpretability in molecular learning

While machine learning models, including deep generative models, have been successful in deriving an informative representation of different classes of molecules, it remains non-trivial and largely unexplored to infer the relationship between different physicochemical and functional attributes of the data. Though laws of chemistry and physics offer broad knowledge on that relationship, those are not enough to establish the (causal) mechanisms active in the system. For that reason, most experimental studies deal with partially known causal relationships, while confounding and observational bias factors (e.g., different experimental conditions such as temperature, assays, solution buffer) are abundant. In this direction, recently Ziatdinov et al. (2020) have established the pairwise causal directions between a set of data descriptors of the microscopic images of molecular surfaces. Interestingly, they found that the causal relationships are consistent across a range of molecular composition.

To our knowledge, this is the first work that infers a graphical model between data attributes from the latents of a generative model of molecules and shows the generalizability of the inferred model across different data distributions.

# 3 Algorithm

## 3.1 Problem setup

Consider a generative model consisting of a decoder $Dec$ that takes input latent code $z \in \mathbb{R}^{d \times 1}$ and generates a data point $x = Dec(z) \in \mathbb{X}^{l \times 1}$ and an encoder $Enc$ that takes a data point and embeds it in the latent space $z = Enc(x)$. We assume that it has been pre-trained using some training method (e.g., Kingma & Welling (2014); Makhzani et al. (2015)) on the data distribution $\mathbb{P}(x)$ sampled from the domain $\mathcal{X}$. We further have access to a vector of attributes $\mathbf{a}(x) \in \mathbb{R}^{|A| \times 1}$ as metadata along with datapoints $x$. In most cases, these metadata attributes can be directly computed on $x$, i.e., $a_i(x) : \mathbb{X}^{l \times 1} \to \mathbb{R}$. For those attributes we cannot directly compute on $x$, we assume that we also have access to an attribute estimator trained on the data $z$ from $p_{a_i}(\cdot) : \mathbb{R}^{d \times 1} \to \mathbb{R}$ producing a regression estimate for attribute $a_i(x) = p_{a_i}(Enc(x))$.

The key problem we would like to solve is to find the structure of the graphical model implied by the generative model and attribute classifier combination, where latent codes $z$ act as the exogenous variables while $\mathbf{a}(x)$ act as the observed variables. In other words, we hypothesize a structural equation model as follows:

$$a_i(x) = f_i(\mathbf{a}_{\mathrm{Pa}(i)}(x), z_{k_i}), \ \forall i \in [1 : |A|] \tag{1}$$

Here, $\mathbf{a}_{\mathrm{Pa}(i)}$ is a subset of the attributes $\mathbf{a}(x) = a_1(x) \dots a_{|A|}(x)$ that form the parents of $a_i(x)$. Also, note that the 'exogenous' variables (i.e. $z_{k_i} \in \{z_1 \dots z_d\}$) are actually the latent representations/codes at the input of the generator. We call $z_{k_i}$ the latent variable associated with attribute $a_i(\cdot)$. Together, the map from the space of latents $z$ to $\mathbf{a}(x)$ can be called the probabilistic mechanism $\mathbf{a}(x) = M(z)$ as represented by the graphical model given by $\{f_1(\cdot) \dots f_{|A|}(\cdot)\}$.

We can define a DAG $G(A, E)$ where $(i, j) \in E$ if $i \in \mathrm{Pa}(j)$ in the structural equation model. Because it is dependent on the latent representation $z$, we call this DAG the "structure of the graphical model implied by the generative model," or simply, the *attribute graph*. This means it may actually improve as state-of-the-art generative models improve. Our principal aim is to learn an attribute graph that is consistent with perturbation experiments on the latent codes $z_i$. The attribute graph should thus be viewed as a summary of how changes to the learnt latent components translate to changes in the predicted/estimated attributes,

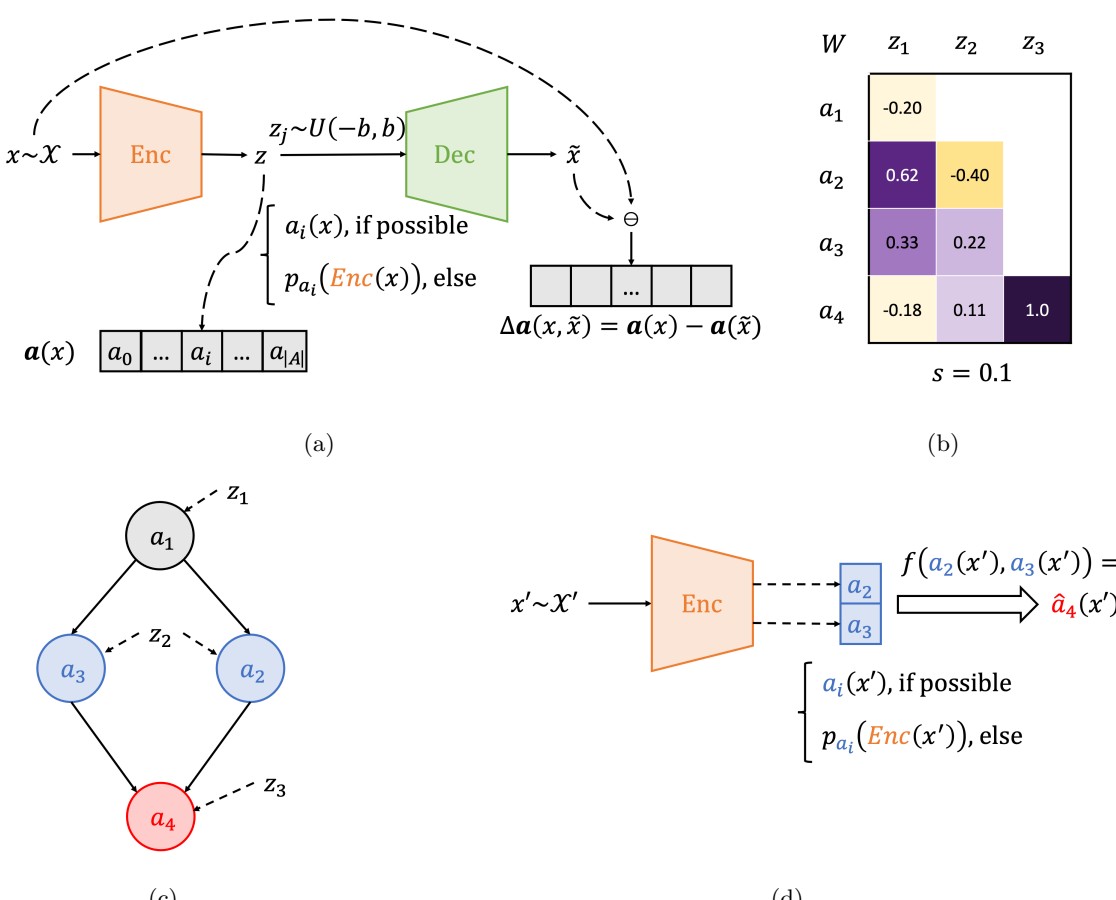

Figure 1: PerturbLearn overview illustrating the: (a) perturbation procedure involving pre-trained generative model $Enc(\cdot), Dec(\cdot)$ and attributes $\mathbf{a}(x)$ resulting in $\Delta\mathbf{a}(x, \tilde{x})$, (b) weight matrix heatmap derivation from perturbations (sparsity threshold $s = 0.1$), (c) DAG building from sparse weights (red indicates the attribute of interest and blue indicates the Markov blanket), (d) application of function $f$ learned on Markov blanket features to OOD samples from $\mathcal{X}'$.

assuming these changes are mostly mediated through influences among the attributes, rather than through a direct influence from latents on attributes.

We then investigate how the attribute graph implied by the generative autoencoder can help train a predictor for a given attribute from other attributes, which can generalize on OOD data. Suppose we want to predict the property $a_i(x)$, then, using the DAG, we take the Markov blanket of $a_i$, i.e., the set of attributes $\text{MB}(a_i) \subset A \cup Z$ which makes it independent of the rest of the variables. This consists of all parents, children, and co-parents in a DAG. Since this subset contains only the features relevant to predicting $a_i$, we should end up with a robust model of minimal size. Furthermore, this feature selection is important for explainability since it is justified by Bayesian reasoning. We assume the structure of the graph is valid across both domains — only the edge weights might change which we can easily fine-tune with minimal samples necessary from the new domain. We visualize PerturbLearn, our solution to this problem, in Figure 1.

## 3.2 PerturbLearn algorithm overview

Our key idea is the following observation, if we take $z_{k_i}$ associated with attribute $a_i(x)$, perturbing it to $\tilde{z}_{k_i}$ would actually affect $a_i$ and all its descendants in the attribute graph. So, we first obtain the sparse perturbation map between each latent $z_j$ and the subset of attributes $A_j \subset A$ that it influences upon

perturbation. Then, we apply a *peeling* algorithm that would actually find the attribute that is associated with a specific latent. In other words, we would find that attribute that occurs last in the ancestral order that is influenced by that latent variable. We use this algorithm to find a DAG with minimal edges that is consistent with the attribute sets for every latent variable. These steps are described in more detail below.

### 3.2.1 Perturbation procedure

Given the pre-trained generative model $Enc(\cdot), Dec(\cdot)$ and a set of property predictors $\mathbf{a}(\cdot)$, we:

1. Encode the sequence $x$ into the latent code $z$ through the encoder $Enc(\cdot)$,

2. Choose any single dimension $j$ in the latent space $z$ and change $z_j$ to $\tilde{z}_j$ that is uniformly sampled in $[-b, b]$ (approximate range of the latent variables when encoded from the data points),

3. Obtain $\tilde{x} = Dec(\tilde{z})$,

4. Estimate the value of all attributes $\mathbf{a}(\tilde{x})$. Note: any attribute which relies on a predictor $p_{a_i}(\cdot)$ will use the matching $Enc(\tilde{x}) = Enc(Dec(\tilde{z}))$. The attribute space is effectively discrete — it only makes sense to discuss attributes of *sequences* (or encodings thereof) — so not every perturbed $\tilde{z}$ will refer to a new sequence (and therefore, attributes).

5. Obtain the net influence vector $\Delta\mathbf{a}(x, \tilde{x}) = \mathbf{a}(x) - \mathbf{a}(\tilde{x})$. Influence values are then standardized (zero mean, unit variance) to scale the influences since attributes may have vastly different ranges. This in turn leads to weights which can be compared against each other.

For the purposes of this work, we assume a linear relationship between latents and attributes although the strength need not be precise, as explained later. Therefore, we then learn a linear model (OLS) to predict the change in attributes $\Delta\mathbf{a}(x, \tilde{x})$ from the latent perturbations $z_j - \tilde{z}_j$ for all data points $x$. This is repeated for every latent dimension $j$. We obtain a weight matrix $W \in \mathbb{R}^{|A| \times |Z|}$ relating attributes in the rows to latent variables in the columns. If we pick the elements whose absolute weights are above a specific threshold $s$, we obtain a sparse matrix. In theory, any sparse linear coefficient learning method would work here (e.g., LASSO (Tibshirani, 1996)) however, in practice, due to the number of samples necessary, the computational load is intractable without the ability to parallelize the computation per latent dimension as in OLS. At this point, the values are simply binary — influenced or not. This would represent the attribute subsets each latent dimension influences. This is summarized in Algorithm 1. Now, we need to associate a latent to one or more attributes such that all attributes influenced by this latent appear later in the ancestral order.

### 3.2.2 Building the DAG

`PerturbLearn` constructs the attribute graph iteratively starting from sink nodes (Algorithm 2). Suppose one could find a latent variable that influences only one attribute and suppose that the graph is a DAG, then that attribute should have no children. Therefore that node is added to the graph (and the corresponding latent is associated with it) (Line 10 in Algorithm 2) and removed from the weight matrix (Lines 17–18 in Algorithm 2). Sometimes, during this recursion, there may be no latent variable that may be found to affect a single remaining attribute. We find a subset with smallest influence (Lines 6–7) and add a confounding arrow between those attributes (Lines 11–13) and add it to the graph. Now, if this latent affects any other downstream nodes previously removed, then we draw an edge from the current set of attributes to them (Lines 14–15). In other words, we assume influences can be explained via mediation through attributes rather than direct influence from the latents. After the recursive procedure is performed, we obtain the transitive reduction (Aho et al., 1972) of the resulting DAG (Line 20). It is the minimal unique edge subgraph that preserves all ancestral relations. This would be the minimal graphical model that would still preserve all latent-attribute influence relationships. See Figure 2 for a visualization of this process on a toy example.

---

**Algorithm 1** `PerturbLearn` — Perturbations to influence weights

---

**Require:** data $X$, attribute estimators $\mathbf{a}$, sample size $n$, number of perturbations $p$, sampling bounds $b$, sparsity $0 < s \leq 1$

1: $\Delta Z, \Delta A = [\ ]$        ▷ empty arrays
2: **for** $j \in 0 \dots |Z|$ **do**
3:      **for** $x_i \in \text{sample}(X, n)$ **do**
4:          $z = \tilde{z} = Enc(x_i)$
5:          **loop** $p$ times
6:              $\tilde{z}_j \sim U(-b, b)$
7:              $\tilde{x} = Dec(\tilde{z})$
8:              append $\Delta z = z - \tilde{z}$ to $\Delta Z$        ▷ net latent vector
9:              append $\Delta\mathbf{a} = \mathbf{a}(x) - \mathbf{a}(\tilde{x})$ to $\Delta A$        ▷ net influence vector
10:          **end loop**
11:      **end for**
12: **end for**
13: $\Delta A = \text{standardize}(\Delta A)$        ▷ center columns to zero mean and scale to unit variance
14: $W = OLS(\Delta Z, \Delta A)$        ▷ least squares weights predicting $\Delta\mathbf{a}$ from $\Delta z_j$
15: $W[\text{abs}(W_{i,j}) \leq s] = 0$        ▷ sparsify weights
16: **return** $W$

---

**Algorithm 2** `PerturbLearn` — Sparse weights to attribute graph

---

**Require:** attributes $A$, latent features $Z$, weights $W \in \mathbb{R}^{|A| \times |Z|}$

1: $I = \{z_j : \{a_i \quad \forall i \in |A| \text{ if } W_{i,j} \neq 0\} \quad \forall j \in |Z|\}$        ▷ influence sets for each latent feature
2: $I_{orig} = I$        ▷ store original influence sets for reference
3: $G =$ empty directed graph
4: $E = [\ ]$        ▷ empty list to store confounded pairs of attributes
5: **while** $I$ is not empty **do**
6:      $N = [|I[z_j]| \quad \forall z_j \in I]$        ▷ count number of attributes influenced by each latent feature
7:      $S = \{z_j : I[z_j] \quad \forall z_j \in \arg\min N\}$        ▷ select subsets of minimal influence
8:      **for** $z_p \in S$ **do**
9:          $P = S[z_p]$        ▷ parents
10:          add node$(a_i)$ to G      $\forall a_i \in P$
11:          **if** $|P| > 1$ **then**        ▷ nodes are confounded
12:              append permutations$(P, 2)$ to E        ▷ save all pairs of nodes in $P$ for later
13:          **end if**
14:          $C = I_{orig}[z_p] \setminus I[z_p]$        ▷ children
15:          add edge$(i, j)$ to G      $\forall i \in P, \forall j \in C$        ▷ add edges from parents to children
16:      **end for**
17:      drop $z_p$ from $I$      $\forall z_p \in S$        ▷ remove latents from further steps
18:      drop $a_i$ from $I[z_j]$      $\forall z_p \in S, \forall a_i \in S[z_p]; \quad \forall z_j \in I$        ▷ remove attributes from further steps
19: **end while**
20: $G =$ transitive_reduction$(G)$
21: add edge$(i, j)$      $\forall i, j \in E$        ▷ add confounded edges
22: **return** $G$

---

## 4 Data and models

### 4.1 Datasets

We conduct experiments on nine pharmacokinetic property prediction problems, an area of science where data limitations are prevalent and label acquisition is costly and time-consuming. Acquiring additional data often involves performing physical synthesis and testing of the molecule or detailed computer simulations of

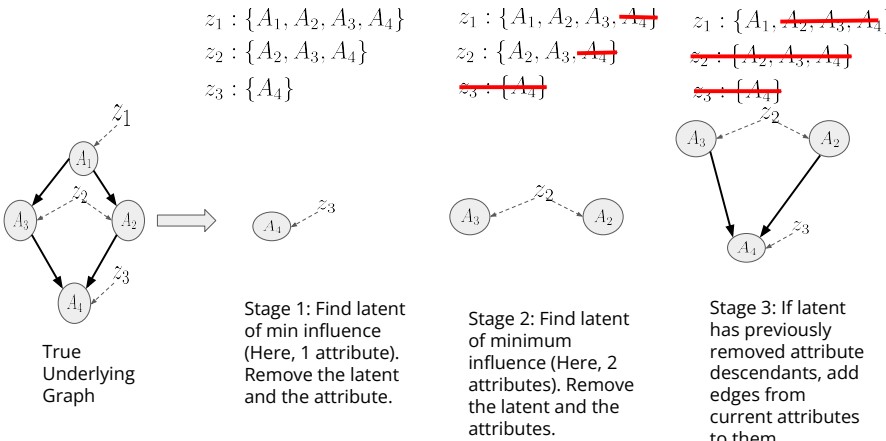

Figure 2: Illustrating various stages of `PerturbLearn` (Algorithm 2) using a toy underlying ground truth attribute model. Stages 1 and 2 correspond to lines 10 and 17–18 in Algorithm 2. Stage 3 corresponds to lines 14–15.

Table 1: Datasets used in experiments along with total size, evaluation metric, and original reference. All datasets were loaded and split according to scaffolds using the Therapeutics Data Commons API.

| Name | Size | Metric | Split | Property | Reference |
|------|------|--------|-------|----------|-----------|
| FreeSolv | 600 | MAE | Scaffold | Absorption | Mobley & Guthrie (2014) |
| Half-life | 667 | Spearman | Scaffold | Excretion | Obach et al. (2008) |
| Caco-2 | 906 | MAE | Scaffold | Absorption | Wang et al. (2016) |
| Hepatocyte | 1020 | Spearman | Scaffold | Excretion | Wenlock & Tomkinson (2015) |
| Microsome | 1102 | Spearman | Scaffold | Excretion | Wenlock & Tomkinson (2015) |
| VDss | 1130 | Spearman | Scaffold | Distribution | Lombardo & Jing (2016) |
| PPBR | 1797 | MAE | Scaffold | Distribution | Wenlock & Tomkinson (2015) |
| Lipophilicity | 4200 | MAE | Scaffold | Absorption | Wenlock & Tomkinson (2015) |
| Solubility | 9982 | MAE | Scaffold | Absorption | Sorkun et al. (2019) |

the system, which demand time, computational resources, and domain experts. Using existing data more effectively from similar domains is one way to lower barrier costs to solving problems in these areas.

Therapeutics Data Commons (TDC) is a platform for AI-powered drug discovery which contains a number of datasets with prediction tasks for drug-relevant properties (Huang et al., 2021). We use all of the regression tasks in the pharmacokinetics domain – i.e., drug absorption, distribution, metabolism, and excretion (ADME). These are summarized in Table 1. All datasets were divided into training, validation, and testing splits according to a ratio of 70%, 10%, and 20%, respectively, based on scaffold (the core structure of the molecule) in order to separate structurally distinct compounds. Because the structures differ greatly, unlike in a random split, scaffold splitting approximates a realistic distribution shift and is widely used to measure robust generalization in ADME prediction (Huang et al., 2021). All datasets were also filtered to include only organic molecules (atoms $\subseteq$ {B, C, N, O, F, P, S, Cl, Br, I}) with additional salt ions removed and pre-processed to use a canonical form of SMILES without isomeric information.

## 4.2 Data attributes

We use a set of molecular descriptors from RDKit as features for the experiments. A full list of the 207 descriptors can be found in the Table 12 (note: `Ipc` was not used due to numerical instabilities in the implementation).

### 4.3 Generative autoencoder details

For small molecule representation, we use the VAE from Chenthamarakshan et al. (2020). This model was trained primarily on the MOSES dataset (Polykovskiy et al., 2020) which is a subset of the ZINC Clean Leads dataset (Irwin et al., 2012). These are 1.6M molecules which are considered lead-like for drug development with benign functionality. The encoder uses a gated recurrent unit (GRU) with a linear output layer while the decoder is a 3-layer GRU with dropout. The encoded latent distribution is constrained to be close to a diagonal Gaussian Distribution, i.e., $q_\phi(z|x) = N(z; \mu(x), \Sigma(x))$ where $\Sigma(x) = \text{diag}[\exp(\log(\sigma_d^2)(x))]$. The model was first trained unsupervised, followed by training jointly with a set of attribute regressors to encourage disentanglement in the learnt space. We use the original model as-is, without any fine-tuning.

## 5 Methods

In order to validate the attribute graphs learned on the latent features of the generative autoencoder, we devise a series of experiments for learning with limited data. If the graph is valid, the Markov blanket of a given node should include all the features needed to predict that attribute and only those features. We hypothesize that this will lead to improved generalization and robustness when learning on only a few data points. We assume that the structure of the attribute graph will not change when the domain shifts, although the relative strength of the links may change. However, these functions should be easy to learn given the minimal set of independent variables and therefore only a small dataset is needed.

To this end, we consider a scenario in which we wish to train a regressor to predict a certain value of interest in a niche domain. There is limited data available for training in this domain, however, data from similar domains are more plentiful. This is a realistic scenario for many real-world applications where the cost of acquiring more data is high and throughput is low due to the involvement of physical experimentation, slow virtual simulation, or laborous user feedback. Therefore, we opt to leverage the larger dataset for pre-training the regressor and the smaller dataset for fine-tuning. With our proposed method, we also use the larger dataset to learn an attribute graph from which we can extract the Markov blanket features.

For every experiment in this work, we take 10 perturbations of 2500 samples (sampled with replacement from the training data) for each dimension in the latent space. We also train a target attribute estimator using the latent representations of the data which is used to estimate the influence of perturbed samples. We then apply PerturbLearn using these inputs along with the pre-trained generative autoencoder to obtain our attribute graph modeling the original domain.

We test our method against five baseline sets of features. Each baseline uses the same model described in the next paragraph. The first baseline uses the full latent vector, $z$, as input features to the predictor model. The second baseline uses all available attributes (i.e., all possible nodes in the attribute graph). The third baseline is simply a concatenation of the first two (all attributes $+z$). We also compare our model trained on the features from the Markov blanket of the attribute of interest to the blankets from causal graphs derived using Greedy Equivalence Search (GES) (Chickering, 2002) and LiNGAM (Shimizu et al., 2006) as well as the feature sets chosen by Sparse Group Lasso (Simon et al., 2013) and Predictive Permutation Feature Selection (PPFS) (Hassan et al., 2021) learned on the training data split. GES and LiNGAM are causal discovery methods which learn based on observational (labeled) data. Group Lasso is a form of structured sparsity regularization which aims to learn a reduced set of input features used for prediction while PPFS is a Markov blanket-based feature-selection algorithm. Finally, we demonstrate that these methods outperform a naïve baseline which always predicts the training mean (dummy).

For initial (base model) training, we use a multilayer perceptron (MLP) model architecture for each feature set for comparison. The MLP hyperparameters are tuned using 5-fold cross-validation on the training set where the search space is a grid of combinations of: hidden layer size 64, 128, or 512 (2 hidden layers chosen independently); dropout rate 0.25 or 0.5; and training duration 100 or 500 epochs. All models use a mean squared error (MSE) loss with a batch size of 256 (or the size of the dataset, if smaller), rectified linear unit (ReLU) activations, and Adam optimization with a learning rate of 0.001. Data is also scaled to zero mean and unit variance independently for each feature. For the Half-life and VDss datasets, we scale the targets in the training set by taking the natural logarithm before learning — the outputs are exponentiated

Table 2: Mean absolute error (MAE $\pm \sigma$; lower is better) of different predictors (with input dimension "size") on the FreeSolv benchmark with varying numbers of test samples included in fine-tuning ($n$). Best results in bold; second-best is underlined.

|  | size | $n = 0$ | $n = 7$ | $n = 10$ | $n = 25$ | $n = 50$ | $n = 100$ |
|---|---|---|---|---|---|---|---|
| dummy | 0 | $2.92_{\pm 0.00}$ | $2.75_{\pm 0.03}$ | $2.71_{\pm 0.01}$ | $2.65_{\pm 0.01}$ | $2.62_{\pm 0.01}$ | $2.62_{\pm 0.02}$ |
| $z$ | 128 | $1.33_{\pm 0.04}$ | $1.70_{\pm 0.06}$ | $1.50_{\pm 0.06}$ | $1.22_{\pm 0.05}$ | $1.10_{\pm 0.05}$ | $0.99_{\pm 0.05}$ |
| all attributes | 208 | $1.22_{\pm 0.09}$ | $1.49_{\pm 0.06}$ | $1.29_{\pm 0.07}$ | $1.04_{\pm 0.05}$ | $0.92_{\pm 0.04}$ | $0.80_{\pm 0.04}$ |
| all attributes$+z$ | 336 | $1.36_{\pm 0.07}$ | $1.60_{\pm 0.08}$ | $1.36_{\pm 0.04}$ | $1.09_{\pm 0.03}$ | $0.96_{\pm 0.04}$ | $0.82_{\pm 0.04}$ |
| GES blanket | 154 | $1.13_{\pm 0.05}$ | $1.50_{\pm 0.06}$ | $1.33_{\pm 0.07}$ | $1.06_{\pm 0.05}$ | $0.93_{\pm 0.04}$ | $0.84_{\pm 0.03}$ |
| LiNGAM blanket | 174 | $1.16_{\pm 0.05}$ | $1.48_{\pm 0.07}$ | $1.31_{\pm 0.04}$ | $1.04_{\pm 0.03}$ | $0.92_{\pm 0.03}$ | $0.81_{\pm 0.03}$ |
| Group Lasso blanket | 103 | $1.15_{\pm 0.04}$ | $\underline{1.36}_{\pm 0.03}$ | $\underline{1.18}_{\pm 0.04}$ | $\mathbf{0.94}_{\pm 0.03}$ | $\mathbf{0.83}_{\pm 0.03}$ | $\mathbf{0.73}_{\pm 0.04}$ |
| PPFS blanket | 121 | $\underline{1.04}_{\pm 0.03}$ | $1.39_{\pm 0.05}$ | $1.19_{\pm 0.03}$ | $0.95_{\pm 0.02}$ | $0.85_{\pm 0.02}$ | $0.76_{\pm 0.03}$ |
| PL blanket | 174 | $\mathbf{0.99}_{\pm 0.07}$ | $\mathbf{1.33}_{\pm 0.07}$ | $\mathbf{1.16}_{\pm 0.06}$ | $\underline{0.95}_{\pm 0.04}$ | $\underline{0.85}_{\pm 0.03}$ | $\underline{0.75}_{\pm 0.02}$ |

before measuring performance metrics. Note, the base model for the $z$-baseline is effectively equivalent to the target attribute estimator used in the PerturbLearn step to make predictions for generated samples.

Fine-tuning on the second domain uses the MLP base model as a feature extractor by freezing the weights and using the outputs from the last hidden layer. These weights are then fed into a Gaussian Process (GP) regressor with a kernel consisting of a sum of two radial basis function (RBF) kernels and a white noise kernel. We optimize the kernel parameters with the Broyden–Fletcher–Goldfarb–Shanno (L-BFGS-B) method with 50 restarts. In order to get a robust estimate of the performance of the model, we run each fine-tuning experiment 10 times on randomly drawn subsets and take the mean. We also repeat the entire procedure 8 times (retraining the base model) to obtain the mean and standard deviation values in Tables 2–11.

For all the experiments, we utilize the provided scaffold splits from TDC. For each of the tasks, we perform graph learning with PerturbLearn and initial regressor training using the "train" split and report the fine-tuned results on the "test" set. We treat the "validation" and "test" sets as effectively two independent shifted domains. In all cases we use the full split when training (and graph learning) but restrict the testing samples to set sizes.

When applying our method, PerturbLearn, after learning the linear weights from the perturbation data, we must choose a sparsity threshold before converting the data to a DAG. We tune this hyperparameter by choosing the best performing threshold on the validation set for the smallest fine-tuning size, $n$. The sparsity threshold is chosen from the set, $\{0.005, 0.01, 0.025, 0.05, 0.075, 0.1, 0.2, 0.25\}$.

For all experiments, we use machines with Intel Xeon Gold 6258R CPUs and NVIDIA Tesla V100 GPUs with up to 32 GB of RAM.

# 6 Results

Table 2 shows the results of experiments using the FreeSolv dataset. Each row shows mean absolute error (MAE) results (mean $\pm$ standard deviation ($\sigma$)) averaged over 10 runs of fine-tuning the MLP base model (learned from the "train" split) with a GP regressor using either the PerturbLearn Markov blanket (also learned from the "train" split) or various baselines while each column shows a different number of fine-tuning samples, $n$, from the "test" split. The $n = 0$ column shows the performance of the base model, without fine-tuning, on the "test" split. The best performing model on the "validation" split with $n = 7$ is used to choose PerturbLearn hyperparameters.

The performance of our method ("PL blanket") is best for small fine-tuning sizes ($n = 0, 7, 10$) and second-best for larger ones ($n = 25, 50, 100$) for the FreeSolv dataset. The performance of our method on the Half-life dataset (Table 3) shows improvement over all baselines other than PPFS in all scenarios. However, the PPFS algorithm involves aggregating $k$ Markov blankets independently derived from separate folds of the

Table 3: Spearman correlation ($\rho \pm \sigma$; higher is better) of different predictors (with input dimension "size") on the Half-life benchmark with varying numbers of test samples included in fine-tuning ($n$). Best results in bold; second-best is underlined. Note: since the dummy regressor output is constant, the Spearman correlation is undefined for that row.

| | size | $n = 0$ | $n = 7$ | $n = 10$ | $n = 25$ | $n = 50$ | $n = 100$ |
|---|---|---|---|---|---|---|---|
| dummy | 0 | — | — | — | — | — | — |
| $z$ | 128 | $0.27_{\pm 0.04}$ | $0.05_{\pm 0.02}$ | $0.08_{\pm 0.02}$ | $0.13_{\pm 0.02}$ | $0.21_{\pm 0.03}$ | $0.28_{\pm 0.04}$ |
| all attributes | 208 | $0.49_{\pm 0.02}$ | $0.17_{\pm 0.02}$ | $0.21_{\pm 0.02}$ | $0.32_{\pm 0.02}$ | $0.38_{\pm 0.03}$ | $0.42_{\pm 0.03}$ |
| all attributes+$z$ | 336 | $0.44_{\pm 0.02}$ | $0.15_{\pm 0.03}$ | $0.17_{\pm 0.03}$ | $0.28_{\pm 0.04}$ | $0.34_{\pm 0.04}$ | $0.37_{\pm 0.04}$ |
| GES blanket | 11 | $0.31_{\pm 0.03}$ | $0.04_{\pm 0.02}$ | $0.05_{\pm 0.02}$ | $0.07_{\pm 0.02}$ | $0.11_{\pm 0.03}$ | $0.22_{\pm 0.07}$ |
| LiNGAM blanket | 194 | $0.49_{\pm 0.02}$ | $0.17_{\pm 0.01}$ | $0.20_{\pm 0.02}$ | $0.31_{\pm 0.02}$ | $0.37_{\pm 0.02}$ | $0.41_{\pm 0.04}$ |
| Group Lasso blanket | 178 | $0.51_{\pm 0.02}$ | $0.16_{\pm 0.02}$ | $0.22_{\pm 0.04}$ | $0.33_{\pm 0.05}$ | $0.40_{\pm 0.04}$ | $0.44_{\pm 0.03}$ |
| PPFS blanket | 58 | $\mathbf{0.56}_{\pm 0.02}$ | $\mathbf{0.21}_{\pm 0.02}$ | $\mathbf{0.26}_{\pm 0.02}$ | $\mathbf{0.40}_{\pm 0.01}$ | $\mathbf{0.47}_{\pm 0.02}$ | $\mathbf{0.49}_{\pm 0.02}$ |
| PL blanket | 109 | $\underline{0.53}_{\pm 0.02}$ | $\underline{0.20}_{\pm 0.03}$ | $\underline{0.24}_{\pm 0.02}$ | $\underline{0.35}_{\pm 0.03}$ | $\underline{0.43}_{\pm 0.03}$ | $\underline{0.47}_{\pm 0.03}$ |
| PL blanket (FS) | 174 | $0.48_{\pm 0.02}$ | $0.15_{\pm 0.02}$ | $0.19_{\pm 0.02}$ | $0.29_{\pm 0.03}$ | $0.39_{\pm 0.02}$ | $0.46_{\pm 0.02}$ |

Table 4: Fine-tuning results for TDC datasets at $n = 25$. FreeSolv, Caco-2, and PPBR use MAE ($\pm \sigma$) while Half-life, VDss, and the Clearance datasets (Hepato. and Micro.) use Spearman correlation ($\pm \sigma$) as the performance metric. Best result in bold; next-best is underlined.

| | FreeSolv ($\downarrow$) | Half-life ($\uparrow$) | Caco-2 ($\downarrow$) | Hepato. ($\uparrow$) | Micro. ($\uparrow$) | VDss ($\uparrow$) | PPBR ($\downarrow$) |
|---|---|---|---|---|---|---|---|
| dummy | $2.65_{\pm 0.01}$ | — | $0.58_{\pm 0.00}$ | — | — | — | $11.03_{\pm 0.09}$ |
| $z$ | $1.22_{\pm 0.05}$ | $0.13_{\pm 0.02}$ | $0.45_{\pm 0.01}$ | $0.15_{\pm 0.03}$ | $0.23_{\pm 0.03}$ | $0.32_{\pm 0.04}$ | $10.98_{\pm 0.13}$ |
| all attributes | $1.04_{\pm 0.05}$ | $0.32_{\pm 0.02}$ | $\underline{0.42}_{\pm 0.01}$ | $\mathbf{0.22}_{\pm 0.02}$ | $0.47_{\pm 0.02}$ | $\underline{0.60}_{\pm 0.02}$ | $10.27_{\pm 0.12}$ |
| all attributes+$z$ | $1.09_{\pm 0.03}$ | $0.28_{\pm 0.04}$ | $\mathbf{0.41}_{\pm 0.01}$ | $\underline{0.21}_{\pm 0.01}$ | $0.47_{\pm 0.01}$ | $0.55_{\pm 0.02}$ | $\underline{10.26}_{\pm 0.14}$ |
| GES blanket | $1.06_{\pm 0.05}$ | $0.07_{\pm 0.02}$ | $\underline{0.42}_{\pm 0.01}$ | $0.20_{\pm 0.02}$ | $0.45_{\pm 0.01}$ | $\underline{0.60}_{\pm 0.02}$ | $10.88_{\pm 0.12}$ |
| LiNGAM blanket | $1.04_{\pm 0.03}$ | $0.31_{\pm 0.02}$ | $\mathbf{0.41}_{\pm 0.01}$ | $\underline{0.21}_{\pm 0.02}$ | $\mathbf{0.49}_{\pm 0.01}$ | $\underline{0.60}_{\pm 0.01}$ | $10.28_{\pm 0.11}$ |
| Group Lasso blanket | $\mathbf{0.94}_{\pm 0.03}$ | $0.33_{\pm 0.05}$ | $0.45_{\pm 0.01}$ | $0.18_{\pm 0.01}$ | $\mathbf{0.49}_{\pm 0.02}$ | $0.57_{\pm 0.02}$ | $10.45_{\pm 0.19}$ |
| PPFS blanket | $\underline{0.95}_{\pm 0.02}$ | $\mathbf{0.40}_{\pm 0.01}$ | $0.43_{\pm 0.02}$ | $\underline{0.21}_{\pm 0.01}$ | $0.45_{\pm 0.02}$ | $\underline{0.60}_{\pm 0.01}$ | $10.56_{\pm 0.17}$ |
| PL blanket | $\underline{0.95}_{\pm 0.04}$ | $\underline{0.35}_{\pm 0.02}$ | $\mathbf{0.41}_{\pm 0.01}$ | $0.19_{\pm 0.01}$ | $0.47_{\pm 0.01}$ | $0.59_{\pm 0.02}$ | $10.36_{\pm 0.17}$ |
| PL blanket (FS) | | $0.29_{\pm 0.03}$ | $\mathbf{0.41}_{\pm 0.01}$ | $0.20_{\pm 0.02}$ | $\underline{0.48}_{\pm 0.01}$ | $\mathbf{0.62}_{\pm 0.01}$ | $\mathbf{10.16}_{\pm 0.16}$ |

data to handle intrinsic sample inefficiency, whereas PertubLearn operates on a single MB, and is therefore is less complex.

These are also the smallest of all the datasets used in our experiments, each under 700 total samples. For the rest of the datasets (Tables 5–11), our method is competitive with the best baselines, often outperforming or nearly indistinguishable, especially with few fine-tuning samples, however, in general, there is little difference between the performance of the feature sets with sufficient training split size.

In some cases, the performance at $n = 0$ is better than when we include additional fine-tuning samples ($n > 0$). In these cases, the models may benefit from a different method of fine-tuning which causes less forgetting than learning a new model on top of a pre-trained feature encoder, however, in the interest of consistency across datasets we keep the fine-tuning procedure the same for all experiments. Since the baseline methods also experience a proportional drop in performance in these cases, the performance of our method under the fine-tuning procedure presented here still follows the same trends noted above. Other methods of fine-tuning should be explored in future work.

Given the impressive performance of our attribute graph from the FreeSolv experiment, we can also try applying this graph to other tasks. In this case, since the tasks are distinct but within the same family (ADME), we can think of this as a more extreme distribution shift problem. These results can be seen in the "PL blanket (FS)" row of Tables 3–11. In a few cases, such as VDss and PPBR, this leads to a noticeable

improvement, surpassing the best baselines. In the rest, there is little difference compared to the task-specific PerturbLearn blanket.

Table 4 shows fine-tuning results at $n = 25$ where we observe in every case, except Hepatocyte clearance, either the task-specific or transferred PL blanket performs best or second-best. We do not witness a significant decrease in performance margin until data size increases past this range ($> 1800$ total samples). For larger datasets, such as Lipophilicity and Solubility (Tables 10–11), most predictors achieve equivalent performance.

## 7 Discussion

In this study, we proposed a simple framework to extract a graphical model relating different data attributes from the latents of a pre-trained generative model. The influence of a latent on attributes is used to construct the graph. In this way, our method leverages the unsupervised learning techniques which produce powerful models trained on vastly more data than is available and labeled for any individual task. This also means our method can be extended to use better models in the future, given they encode information in a continuous latent space.

We have provided here a domain generalization method that utilizes latent-mediated influences among molecular attributes and have empirically shown its efficacy to predict molecular attributes by learning from limited out-of-distribution data. We showcased the performance of the proposed method, along with other existing feature selection methods, in predicting molecular properties in a low-data regime. Thus, this study provides a comprehensive evaluation of feature selection methods for molecular learning tasks, as well as offers insights into the derived statistical relations between molecular attributes. Taken together, the results show the use of derived Bayesian information improves over the baseline feature sets, especially when data from the training and/or test domains is limited, demonstrating that extracting the dependency information underlying a pre-trained generative model leads to more robust and generalizable models. These models also use only the meaningful and relevant properties as features meaning the models will be both smaller and more interpretable than their baseline counterparts.

One potential limitation of our framework is the dependency on a pre-determined set of data attributes, which may not be comprehensive (or may already involve experts applying implicit causal models learned from experience to choose attributes). Further investigation on the accordance of the derived attribute graph with domain priors, using only low-level and cheaply computable attributes, and/or with expert knowledge would be addressed in future.

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
