# A Appendix

## A.1 Additional results

Table 5: Mean absolute error (MAE $\pm \sigma$; lower is better) of different predictors on the Caco-2 benchmark with varying numbers of test samples included in fine-tuning ($n$). Best results in bold; next-best is underlined.

|  | size | $n=0$ | $n=7$ | $n=10$ | $n=25$ | $n=50$ | $n=100$ |
|---|---|---|---|---|---|---|---|
| dummy | 0 | $0.57_{\pm 0.00}$ | $0.60_{\pm 0.00}$ | $0.59_{\pm 0.00}$ | $0.58_{\pm 0.00}$ | $0.57_{\pm 0.00}$ | $0.57_{\pm 0.00}$ |
| $z$ | 128 | $0.58_{\pm 0.04}$ | $0.55_{\pm 0.01}$ | $0.52_{\pm 0.01}$ | $0.45_{\pm 0.01}$ | $0.41_{\pm 0.01}$ | $0.37_{\pm 0.01}$ |
| all attributes | 208 | $0.60_{\pm 0.09}$ | $\mathbf{0.53}_{\pm 0.01}$ | $\underline{0.51}_{\pm 0.01}$ | $\underline{0.42}_{\pm 0.01}$ | $\underline{0.37}_{\pm 0.01}$ | $\underline{0.32}_{\pm 0.01}$ |
| all attributes+$z$ | 336 | $0.70_{\pm 0.07}$ | $\underline{0.54}_{\pm 0.01}$ | $\mathbf{0.50}_{\pm 0.01}$ | $\mathbf{0.41}_{\pm 0.01}$ | $\mathbf{0.36}_{\pm 0.01}$ | $\mathbf{0.31}_{\pm 0.01}$ |
| GES blanket | 170 | $0.58_{\pm 0.06}$ | $\mathbf{0.53}_{\pm 0.01}$ | $\mathbf{0.50}_{\pm 0.01}$ | $\underline{0.42}_{\pm 0.01}$ | $\mathbf{0.36}_{\pm 0.01}$ | $\underline{0.32}_{\pm 0.01}$ |
| LiNGAM blanket | 199 | $0.61_{\pm 0.07}$ | $\mathbf{0.53}_{\pm 0.01}$ | $\mathbf{0.50}_{\pm 0.01}$ | $\mathbf{0.41}_{\pm 0.01}$ | $\mathbf{0.36}_{\pm 0.01}$ | $\mathbf{0.31}_{\pm 0.01}$ |
| Group Lasso blanket | 62 | $0.68_{\pm 0.05}$ | $\underline{0.54}_{\pm 0.01}$ | $0.52_{\pm 0.01}$ | $0.45_{\pm 0.01}$ | $0.40_{\pm 0.01}$ | $0.34_{\pm 0.01}$ |
| PPFS blanket | 107 | $0.71_{\pm 0.06}$ | $\underline{0.54}_{\pm 0.01}$ | $0.52_{\pm 0.01}$ | $0.43_{\pm 0.02}$ | $0.38_{\pm 0.02}$ | $\underline{0.32}_{\pm 0.01}$ |
| PL blanket | 176 | $\underline{0.56}_{\pm 0.08}$ | $\mathbf{0.53}_{\pm 0.01}$ | $\mathbf{0.50}_{\pm 0.01}$ | $\mathbf{0.41}_{\pm 0.01}$ | $\mathbf{0.36}_{\pm 0.01}$ | $\underline{0.32}_{\pm 0.01}$ |
| PL blanket (FS) | 174 | $\mathbf{0.53}_{\pm 0.04}$ | $\mathbf{0.53}_{\pm 0.01}$ | $\mathbf{0.50}_{\pm 0.01}$ | $\mathbf{0.41}_{\pm 0.01}$ | $\mathbf{0.36}_{\pm 0.01}$ | $\mathbf{0.31}_{\pm 0.01}$ |

Table 6: Spearman correlation ($\rho \pm \sigma$; higher is better) of different predictors on the Hepatocyte clearance benchmark with varying numbers of test samples included in fine-tuning ($n$). Best results in bold; next-best is underlined.

|  | size | $n=0$ | $n=7$ | $n=10$ | $n=25$ | $n=50$ | $n=100$ |
|---|---|---|---|---|---|---|---|
| dummy | 0 | — | — | — | — | — | — |
| $z$ | 128 | $0.27_{\pm 0.01}$ | $0.09_{\pm 0.02}$ | $0.11_{\pm 0.02}$ | $0.15_{\pm 0.03}$ | $0.20_{\pm 0.03}$ | $0.24_{\pm 0.03}$ |
| all attributes | 208 | $\mathbf{0.37}_{\pm 0.01}$ | $\mathbf{0.13}_{\pm 0.01}$ | $\mathbf{0.16}_{\pm 0.01}$ | $\mathbf{0.22}_{\pm 0.02}$ | $\mathbf{0.27}_{\pm 0.02}$ | $0.30_{\pm 0.02}$ |
| all attributes+$z$ | 336 | $0.35_{\pm 0.01}$ | $\mathbf{0.13}_{\pm 0.02}$ | $0.14_{\pm 0.01}$ | $\underline{0.21}_{\pm 0.01}$ | $0.25_{\pm 0.02}$ | $0.29_{\pm 0.02}$ |
| GES blanket | 163 | $0.35_{\pm 0.01}$ | $\underline{0.11}_{\pm 0.02}$ | $\underline{0.15}_{\pm 0.02}$ | $0.20_{\pm 0.02}$ | $0.25_{\pm 0.02}$ | $0.30_{\pm 0.02}$ |
| LiNGAM blanket | 194 | $\mathbf{0.37}_{\pm 0.01}$ | $\mathbf{0.13}_{\pm 0.02}$ | $\mathbf{0.16}_{\pm 0.01}$ | $\underline{0.21}_{\pm 0.02}$ | $\underline{0.26}_{\pm 0.02}$ | $\underline{0.31}_{\pm 0.02}$ |
| Group Lasso blanket | 168 | $0.35_{\pm 0.01}$ | $\underline{0.11}_{\pm 0.02}$ | $0.12_{\pm 0.02}$ | $0.18_{\pm 0.01}$ | $0.23_{\pm 0.02}$ | $0.29_{\pm 0.02}$ |
| PPFS blanket | 107 | $\underline{0.36}_{\pm 0.01}$ | $\mathbf{0.13}_{\pm 0.02}$ | $\mathbf{0.16}_{\pm 0.01}$ | $\underline{0.21}_{\pm 0.01}$ | $\mathbf{0.27}_{\pm 0.01}$ | $\mathbf{0.34}_{\pm 0.01}$ |
| PL blanket | 151 | $\underline{0.36}_{\pm 0.01}$ | $\underline{0.11}_{\pm 0.02}$ | $0.14_{\pm 0.01}$ | $0.19_{\pm 0.01}$ | $0.23_{\pm 0.01}$ | $0.27_{\pm 0.02}$ |
| PL blanket (FS) | 174 | $0.35_{\pm 0.00}$ | $\underline{0.11}_{\pm 0.02}$ | $0.14_{\pm 0.01}$ | $0.20_{\pm 0.02}$ | $0.24_{\pm 0.02}$ | $0.28_{\pm 0.01}$ |

Table 7: Spearman correlation ($\rho \pm \sigma$; higher is better) of different predictors on the Microsome clearance benchmark with varying numbers of test samples included in fine-tuning ($n$). Best results in bold; next-best is underlined.

| | size | $n = 0$ | $n = 7$ | $n = 10$ | $n = 25$ | $n = 50$ | $n = 100$ |
|---|---|---|---|---|---|---|---|
| dummy | 0 | — | — | — | — | — | — |
| $z$ | 128 | $0.43_{\pm 0.02}$ | $0.16_{\pm 0.02}$ | $0.17_{\pm 0.03}$ | $0.23_{\pm 0.03}$ | $0.28_{\pm 0.03}$ | $0.33_{\pm 0.03}$ |
| all attributes | 208 | $\underline{0.65}_{\pm 0.01}$ | $0.39_{\pm 0.04}$ | $\underline{0.43}_{\pm 0.02}$ | $0.47_{\pm 0.02}$ | $0.50_{\pm 0.02}$ | $0.53_{\pm 0.02}$ |
| all attributes+$z$ | 336 | $\mathbf{0.66}_{\pm 0.01}$ | $0.39_{\pm 0.03}$ | $0.42_{\pm 0.02}$ | $0.47_{\pm 0.01}$ | $0.51_{\pm 0.02}$ | $\underline{0.54}_{\pm 0.01}$ |
| GES blanket | 154 | $0.64_{\pm 0.01}$ | $0.37_{\pm 0.02}$ | $0.41_{\pm 0.03}$ | $0.45_{\pm 0.01}$ | $0.49_{\pm 0.02}$ | $0.51_{\pm 0.03}$ |
| LiNGAM blanket | 191 | $\mathbf{0.66}_{\pm 0.01}$ | $\mathbf{0.41}_{\pm 0.02}$ | $\underline{0.43}_{\pm 0.02}$ | $\mathbf{0.49}_{\pm 0.01}$ | $\mathbf{0.53}_{\pm 0.01}$ | $\mathbf{0.55}_{\pm 0.02}$ |
| Group Lasso blanket | 172 | $\mathbf{0.66}_{\pm 0.01}$ | $0.39_{\pm 0.03}$ | $\mathbf{0.44}_{\pm 0.02}$ | $\mathbf{0.49}_{\pm 0.02}$ | $\underline{0.52}_{\pm 0.02}$ | $\mathbf{0.55}_{\pm 0.03}$ |
| PPFS blanket | 101 | $0.62_{\pm 0.02}$ | $0.35_{\pm 0.02}$ | $0.40_{\pm 0.01}$ | $0.45_{\pm 0.02}$ | $0.49_{\pm 0.02}$ | $0.52_{\pm 0.02}$ |
| PL blanket | 121 | $\underline{0.65}_{\pm 0.01}$ | $\underline{0.40}_{\pm 0.02}$ | $\mathbf{0.44}_{\pm 0.01}$ | $0.47_{\pm 0.01}$ | $0.50_{\pm 0.01}$ | $0.53_{\pm 0.02}$ |
| PL blanket (FS) | 174 | $\mathbf{0.66}_{\pm 0.01}$ | $\underline{0.40}_{\pm 0.02}$ | $\underline{0.43}_{\pm 0.02}$ | $\underline{0.48}_{\pm 0.01}$ | $0.51_{\pm 0.02}$ | $0.53_{\pm 0.02}$ |

Table 8: Spearman correlation ($\rho \pm \sigma$; higher is better) of different predictors on the VDss benchmark with varying numbers of test samples included in fine-tuning ($n$). Best results in bold; next-best is underlined.

| | size | $n = 0$ | $n = 7$ | $n = 10$ | $n = 25$ | $n = 50$ | $n = 100$ |
|---|---|---|---|---|---|---|---|
| dummy | 0 | — | — | — | — | — | — |
| $z$ | 128 | $0.49_{\pm 0.02}$ | $0.16_{\pm 0.02}$ | $0.21_{\pm 0.04}$ | $0.32_{\pm 0.04}$ | $0.41_{\pm 0.02}$ | $0.47_{\pm 0.01}$ |
| all attributes | 208 | $\underline{0.68}_{\pm 0.01}$ | $0.37_{\pm 0.03}$ | $0.45_{\pm 0.03}$ | $0.60_{\pm 0.02}$ | $\underline{0.65}_{\pm 0.02}$ | $\underline{0.66}_{\pm 0.01}$ |
| all attributes+$z$ | 336 | $0.64_{\pm 0.01}$ | $0.32_{\pm 0.03}$ | $0.39_{\pm 0.03}$ | $0.55_{\pm 0.02}$ | $0.60_{\pm 0.01}$ | $0.62_{\pm 0.01}$ |
| GES blanket | 170 | $\underline{0.68}_{\pm 0.01}$ | $0.40_{\pm 0.03}$ | $\underline{0.48}_{\pm 0.03}$ | $0.60_{\pm 0.02}$ | $0.64_{\pm 0.01}$ | $\underline{0.66}_{\pm 0.01}$ |
| LiNGAM blanket | 196 | $\underline{0.68}_{\pm 0.01}$ | $0.38_{\pm 0.03}$ | $0.45_{\pm 0.03}$ | $0.60_{\pm 0.01}$ | $0.64_{\pm 0.01}$ | $0.65_{\pm 0.01}$ |
| Group Lasso blanket | 145 | $0.65_{\pm 0.01}$ | $0.36_{\pm 0.02}$ | $0.42_{\pm 0.03}$ | $0.57_{\pm 0.02}$ | $0.62_{\pm 0.01}$ | $0.63_{\pm 0.02}$ |
| PPFS blanket | 83 | $\underline{0.68}_{\pm 0.01}$ | $\underline{0.40}_{\pm 0.01}$ | $0.46_{\pm 0.02}$ | $\underline{0.60}_{\pm 0.01}$ | $0.64_{\pm 0.01}$ | $\underline{0.66}_{\pm 0.01}$ |
| PL blanket | 109 | $0.67_{\pm 0.01}$ | $0.39_{\pm 0.03}$ | $0.46_{\pm 0.04}$ | $0.59_{\pm 0.02}$ | $0.64_{\pm 0.01}$ | $0.65_{\pm 0.01}$ |
| PL blanket (FS) | 174 | $\mathbf{0.70}_{\pm 0.01}$ | $\mathbf{0.41}_{\pm 0.02}$ | $\mathbf{0.49}_{\pm 0.03}$ | $\mathbf{0.62}_{\pm 0.01}$ | $\mathbf{0.66}_{\pm 0.00}$ | $\mathbf{0.67}_{\pm 0.01}$ |

Table 9: Mean absolute error (MAE $\pm \sigma$; lower is better) of different predictors on the PPBR benchmark with varying numbers of test samples included in fine-tuning ($n$). Best results in bold; next-best is underlined.

| | size | $n = 0$ | $n = 7$ | $n = 10$ | $n = 25$ | $n = 50$ | $n = 100$ |
|---|---|---|---|---|---|---|---|
| dummy | 0 | $\mathbf{11.40}_{\pm 0.00}$ | $11.45_{\pm 0.25}$ | $11.31_{\pm 0.20}$ | $11.03_{\pm 0.09}$ | $10.89_{\pm 0.07}$ | $10.84_{\pm 0.03}$ |
| $z$ | 128 | $15.26_{\pm 0.26}$ | $11.64_{\pm 0.28}$ | $11.46_{\pm 0.27}$ | $10.98_{\pm 0.13}$ | $10.78_{\pm 0.05}$ | $10.51_{\pm 0.10}$ |
| all attributes | 208 | $15.55_{\pm 0.46}$ | $11.23_{\pm 0.22}$ | $\mathbf{10.92}_{\pm 0.23}$ | $10.27_{\pm 0.12}$ | $9.83_{\pm 0.10}$ | $9.43_{\pm 0.12}$ |
| all attributes+$z$ | 336 | $\underline{14.27}_{\pm 0.33}$ | $\mathbf{11.14}_{\pm 0.21}$ | $\underline{10.96}_{\pm 0.17}$ | $\underline{10.26}_{\pm 0.14}$ | $\underline{9.76}_{\pm 0.10}$ | $\underline{9.30}_{\pm 0.08}$ |
| GES blanket | 12 | $17.44_{\pm 0.39}$ | $11.64_{\pm 0.31}$ | $11.33_{\pm 0.25}$ | $10.88_{\pm 0.12}$ | $10.51_{\pm 0.12}$ | $10.02_{\pm 0.17}$ |
| LiNGAM blanket | 192 | $15.39_{\pm 0.30}$ | $11.25_{\pm 0.11}$ | $11.07_{\pm 0.20}$ | $10.28_{\pm 0.11}$ | $9.79_{\pm 0.14}$ | $9.41_{\pm 0.15}$ |
| Group Lasso blanket | 136 | $16.17_{\pm 0.43}$ | $11.39_{\pm 0.19}$ | $11.03_{\pm 0.29}$ | $10.45_{\pm 0.19}$ | $9.91_{\pm 0.12}$ | $9.47_{\pm 0.12}$ |
| PPFS blanket | 97 | $15.73_{\pm 0.63}$ | $11.31_{\pm 0.24}$ | $11.04_{\pm 0.14}$ | $10.56_{\pm 0.17}$ | $10.08_{\pm 0.10}$ | $9.72_{\pm 0.14}$ |
| PL blanket | 173 | $15.39_{\pm 0.61}$ | $\underline{11.17}_{\pm 0.18}$ | $10.99_{\pm 0.21}$ | $10.36_{\pm 0.17}$ | $9.87_{\pm 0.19}$ | $9.55_{\pm 0.21}$ |
| PL blanket (FS) | 174 | $14.55_{\pm 0.67}$ | $11.20_{\pm 0.15}$ | $\mathbf{10.92}_{\pm 0.31}$ | $\mathbf{10.16}_{\pm 0.16}$ | $\mathbf{9.64}_{\pm 0.14}$ | $\mathbf{9.22}_{\pm 0.19}$ |

Table 10: Mean absolute error (MAE $\pm \sigma$; lower is better) of different predictors on the Lipophilicity benchmark with varying numbers of test samples included in fine-tuning ($n$). Best results in bold; next-best is underlined.

| | size | $n = 0$ | $n = 7$ | $n = 10$ | $n = 25$ | $n = 50$ | $n = 100$ |
|---|---|---|---|---|---|---|---|
| dummy | 0 | $0.99_{\pm 0.00}$ | $1.02_{\pm 0.01}$ | $1.00_{\pm 0.01}$ | $0.98_{\pm 0.00}$ | $0.97_{\pm 0.00}$ | $0.97_{\pm 0.00}$ |
| $z$ | 128 | $0.79_{\pm 0.01}$ | $0.97_{\pm 0.01}$ | $0.94_{\pm 0.01}$ | $0.86_{\pm 0.01}$ | $0.81_{\pm 0.01}$ | $0.78_{\pm 0.01}$ |
| all attributes | 208 | $\mathbf{0.62}_{\pm 0.01}$ | $\mathbf{0.85}_{\pm 0.02}$ | $\mathbf{0.79}_{\pm 0.01}$ | $\mathbf{0.67}_{\pm 0.01}$ | $\underline{0.64}_{\pm 0.01}$ | $\mathbf{0.61}_{\pm 0.01}$ |
| all attributes+$z$ | 336 | $\mathbf{0.62}_{\pm 0.01}$ | $\mathbf{0.85}_{\pm 0.02}$ | $\mathbf{0.79}_{\pm 0.01}$ | $\underline{0.68}_{\pm 0.01}$ | $\underline{0.64}_{\pm 0.01}$ | $\mathbf{0.61}_{\pm 0.01}$ |
| GES blanket | 188 | $\mathbf{0.62}_{\pm 0.01}$ | $\mathbf{0.85}_{\pm 0.01}$ | $0.80_{\pm 0.01}$ | $\underline{0.68}_{\pm 0.01}$ | $\underline{0.64}_{\pm 0.01}$ | $\mathbf{0.61}_{\pm 0.01}$ |
| LiNGAM blanket | 197 | $\mathbf{0.62}_{\pm 0.01}$ | $\mathbf{0.85}_{\pm 0.02}$ | $\mathbf{0.79}_{\pm 0.01}$ | $\underline{0.68}_{\pm 0.01}$ | $\mathbf{0.63}_{\pm 0.01}$ | $\mathbf{0.61}_{\pm 0.00}$ |
| Group Lasso blanket | 66 | $0.72_{\pm 0.01}$ | $0.93_{\pm 0.02}$ | $0.89_{\pm 0.02}$ | $0.78_{\pm 0.02}$ | $0.73_{\pm 0.01}$ | $0.70_{\pm 0.01}$ |
| PPFS blanket | 102 | $0.65_{\pm 0.01}$ | $0.89_{\pm 0.02}$ | $0.83_{\pm 0.01}$ | $0.72_{\pm 0.01}$ | $0.67_{\pm 0.01}$ | $0.64_{\pm 0.01}$ |
| PL blanket | 174 | $\underline{0.63}_{\pm 0.01}$ | $\underline{0.86}_{\pm 0.01}$ | $0.81_{\pm 0.01}$ | $\underline{0.68}_{\pm 0.01}$ | $\underline{0.64}_{\pm 0.01}$ | $\underline{0.62}_{\pm 0.01}$ |
| PL blanket (FS) | 174 | $\mathbf{0.62}_{\pm 0.01}$ | $0.87_{\pm 0.02}$ | $\underline{0.80}_{\pm 0.01}$ | $0.69_{\pm 0.01}$ | $0.65_{\pm 0.01}$ | $\underline{0.62}_{\pm 0.00}$ |

Table 11: Mean absolute error (MAE $\pm \sigma$; lower is better) of different predictors on the Solubility benchmark with varying numbers of test samples included in fine-tuning ($n$). Best results in bold; next-best is underlined.

| | size | $n = 0$ | $n = 7$ | $n = 10$ | $n = 25$ | $n = 50$ | $n = 100$ |
|---|---|---|---|---|---|---|---|
| dummy | 0 | $1.89_{\pm 0.00}$ | $1.94_{\pm 0.01}$ | $1.90_{\pm 0.01}$ | $1.85_{\pm 0.01}$ | $1.83_{\pm 0.00}$ | $1.83_{\pm 0.00}$ |
| $z$ | 128 | $1.08_{\pm 0.01}$ | $1.60_{\pm 0.04}$ | $1.48_{\pm 0.02}$ | $1.25_{\pm 0.02}$ | $1.17_{\pm 0.02}$ | $1.11_{\pm 0.01}$ |
| all attributes | 208 | $\mathbf{0.94}_{\pm 0.01}$ | $\mathbf{1.44}_{\pm 0.03}$ | $\underline{1.31}_{\pm 0.03}$ | $1.09_{\pm 0.02}$ | $\mathbf{1.00}_{\pm 0.01}$ | $\underline{0.96}_{\pm 0.01}$ |
| all attributes+$z$ | 336 | $\mathbf{0.94}_{\pm 0.01}$ | $1.45_{\pm 0.03}$ | $\mathbf{1.30}_{\pm 0.02}$ | $\mathbf{1.07}_{\pm 0.02}$ | $\mathbf{1.00}_{\pm 0.01}$ | $\mathbf{0.95}_{\pm 0.01}$ |
| GES blanket | 178 | $\underline{0.95}_{\pm 0.01}$ | $\mathbf{1.44}_{\pm 0.04}$ | $1.32_{\pm 0.03}$ | $1.09_{\pm 0.01}$ | $\underline{1.01}_{\pm 0.01}$ | $\underline{0.96}_{\pm 0.01}$ |
| LiNGAM blanket | 204 | $\mathbf{0.94}_{\pm 0.01}$ | $1.46_{\pm 0.04}$ | $\mathbf{1.30}_{\pm 0.03}$ | $\underline{1.08}_{\pm 0.01}$ | $\mathbf{1.00}_{\pm 0.01}$ | $\underline{0.96}_{\pm 0.01}$ |
| Group Lasso blanket | 88 | $0.99_{\pm 0.01}$ | $1.51_{\pm 0.02}$ | $1.37_{\pm 0.03}$ | $1.14_{\pm 0.01}$ | $1.05_{\pm 0.01}$ | $1.01_{\pm 0.01}$ |
| PPFS blanket | 146 | $\underline{0.95}_{\pm 0.02}$ | $\underline{1.45}_{\pm 0.04}$ | $1.32_{\pm 0.02}$ | $1.09_{\pm 0.02}$ | $\underline{1.01}_{\pm 0.01}$ | $\underline{0.96}_{\pm 0.01}$ |
| PL blanket | 170 | $\underline{0.95}_{\pm 0.01}$ | $\underline{1.45}_{\pm 0.04}$ | $1.32_{\pm 0.03}$ | $1.10_{\pm 0.02}$ | $\underline{1.01}_{\pm 0.01}$ | $0.97_{\pm 0.01}$ |
| PL blanket (FS) | 174 | $0.96_{\pm 0.02}$ | $\mathbf{1.44}_{\pm 0.02}$ | $\underline{1.31}_{\pm 0.03}$ | $1.09_{\pm 0.02}$ | $\underline{1.01}_{\pm 0.01}$ | $\underline{0.96}_{\pm 0.01}$ |

### A.2 Molecular descriptors

Table 12: Descriptors used in small molecule experiments. All descriptors are calculated using RDKit.

| | | |
|---|---|---|
| Gasteiger/Marsili Partial Charges | NOCount | RingCount |
| BalabanJ | NumHAcceptors | FractionCSP3 |
| BertzCT | NumHDonors | NumSpiroAtoms |
| HallKierAlpha | NumHeteroatoms | NumBridgeheadAtoms |
| Kappa1 - Kappa3 | NumRotatableBonds | TPSA |
| Phi | NumValenceElectrons | LabuteASA |
| Chi0, Chi1 | NumAmideBonds | PEOE_VSA1 – PEOE_VSA14 |
| Chi0n – Chi4n | NumAromaticRings | SMR_VSA1 – SMR_VSA10 |
| Chi0v – Chi4v | NumSaturatedRings | SlogP_VSA1 – SlogP_VSA12 |
| MolLogP | NumAliphaticRings | EState_VSA1 – EState_VSA11 |
| MolMR | NumAromaticHeterocycles | VSA_EState1 – VSA_EState10 |
| MolWt | NumSaturatedHeterocycles | MQNs |
| ExactMolWt | NumAliphaticHeterocycles | Topliss fragments |
| HeavyAtomCount | NumAromaticCarbocycles | Autocorr2D |
| HeavyAtomMolWt | NumSaturatedCarbocycles | BCUT2D |
| NHOHCount | NumAliphaticCarbocycles | |