# OpenReview forum: "Attribute Graphs Underlying Molecular Generative Models: Path to Learning with Limited Data"
_TMLR — Accepted by TMLR_

### Review · Reviewer_8DbN · 2024-05-24

**Summary Of Contributions:**

This paper proposes an algorithm to use perturbation experiments on latent codes of a pre-trained generative model to discover a causal graph, which can be used for generalizable prediction on out-of-distribution testing. Specifically, a generative model is pre-trained on a large amount of unlabeled data, and then the domain-speciﬁc causal graph is extracted using a smaller set of data with easily obtained labels. After that, the authors do a sensitivity analysis of each of the data attributes by perturbing one latent dimension at a time and using that sensitivity information to construct the causal graph of attributes. The proposed method is evaluated on molecule generation. Results show that their model is more robust to distribution shifts when transferred or ﬁne-tuned with a few samples from the new distribution.

**Audience:**

Yes

**Claims And Evidence:**

Yes

**Requested Changes:**

1.	Figure 1 needs to be revised. For example, the font size of texts should be larger considering there is enough space. The order of (b)(c)(d) should be changed. (d) misses a lot of information. How does the train and test process obtain a_2 and a_3? What is the task of this application?
2.	For Figure 2, it would be better if the authors could annotate the corresponding lines of Algorithm 1 in each stage.

**Strengths And Weaknesses:**

**Strengths:**

1.	The writing of this paper is much improved compared to the previous version.
2.	Changing the concept from causality to correlation to describe the attribute graph seems more reasonable.

**Weaknesses:**

1.	The figures still need some revisions to be improved.
2.	The proposed method seems to work well for molecular attribute prediction tasks. However, the title seems quite broad and covers all generative models. To avoid misleading readers, it would be good to narrow down the scope to molecules. I am happy to discuss the title with the author and other reviewers.

---

> ### Author Response · Authors · 2024-06-21
>
> > The proposed method seems to work well for molecular attribute prediction tasks. However, the title seems quite broad and covers all generative models. To avoid misleading readers, it would be good to narrow down the scope to molecules. I am happy to discuss the title with the author and other reviewers.
>
> We have added “Molecular” (Generative Models) to the title. We hope this change in the title is enough to narrow down the scope. We are open to discuss it further with the reviewer if needed.
>
> > Figure 1 needs to be revised. For example, the font size of texts should be larger considering there is enough space. The order of (b)(c)(d) should be changed. (d) misses a lot of information. How does the train and test process obtain a_2 and a_3? What is the task of this application?
>
> Figure 1 has been improved, as you suggested. $a_2$ and $a_3$ are obtained via the process $p_{a_i}(Enc(x))$, if necessary, or computed directly from $x$, when possible. The task is predicting $\hat{a}_4$, in this case, but could be any attribute of interest in $G$. We have added this information in the figure caption.
>
> > For Figure 2, it would be better if the authors could annotate the corresponding lines of Algorithm 1 in each stage.
>
> The caption for Figure 2 has been updated with the requested information.

---

### Review · Reviewer_BBhC · 2024-05-26

**Summary Of Contributions:**

This manuscript proposes an approach to learn a graphical model on attributes by using pertained generative models (PerturbLearn), and then use that graphical model for feature selection in prediction tasks.  This idea can improve generalization in semi-supervised or transfer learning tasks by leveraging available generative modeling information to reduce the complexity of predictive models.

**Audience:**

Yes

**Broader Impact Concerns:**

N/A,

**Claims And Evidence:**

Yes

**Requested Changes:**

Needed:

Uncertainty metrics in tables need to be defined.

Give full details on fine-tuning procedure, and explain the decrease in performance that frequently occurs from fine-tuning.

Describe a model selection approach that could be used in practice on the motivating problems.

**Strengths And Weaknesses:**

Strengths:

The model and ideas are clearly written, and the core idea makes intuitive sense.  The performance seems competitive across the board with a reasonable number of relevant comparisons.

Weaknesses:

The fine-tuning performance is surprising, as the models often get worst across the board when fine-tuning.  The fine-tuning method is not described in sufficient detail to evaluate whether it is reasonable.  Please give a more precise mathematical and algorithmic description.  Furthermore, it is mentioned that the hyperparameters are chosen based upon the n=7 scenario with a validation set.  As such, it seems like performance should not appreciably decrease from the n=0 scenario, as it does in both examples given in the main paper and several in the appendix.

It appears a reasonable-sized validation split is used for model selection, which seems to contradict the message that this method is most appropriate when there are few labeled samples available for model training and selection.  Please describe how model selection would be done in practice in the limited sample case.

---

> ### Author Response · Authors · 2024-06-21
>
> > Uncertainty metrics in tables need to be defined.
>
> We have added clarification to all the table captions regarding the $\pm$ standard deviation that provides the measure of uncertainty.
>
> > Give full details on fine-tuning procedure, and explain the decrease in performance that frequently occurs from fine-tuning.
>
> The fine-tuning procedure uses frozen layers from the $n=0$ MLP but the final layer is replaced with a GP. The GP parameters are fully learned on the $n$ new OOD samples, though, not utilizing training samples at all. For distributions where training and test sets are similar, this would appear to harm the fine-tuning performance since the model would benefit from more information carry-over.
>
> We keep the procedure consistent for all experiments but in a real application one could certainly adjust the fine-tuning procedure as additional hyperparameters.
>
> > Describe a model selection approach that could be used in practice on the motivating problems.
>
> In our experiments, the validation split is half the size of the test split. These are treated as three separate domains, where validation and test are limited in size. In practice, the validation set could be drawn from an unrelated domain which is easier to label, a scaffold split of the training set, or cross-validation may be used on the test set.

---

> > ### Comment · Reviewer_BBhC · 2024-07-05
> > **Brief Response**
> >
> > First, I want to note that GPs have been used to refine error prediction under limited data for decades under terminology such as "residual kriging."  I find it surprising that the authors chose this modeling choice rather than doing something that blends the predictions from the N=0 case, as I expect that this is sub-optimal.  I would suggest looking into these ideas for future work.
> >
> > I would suggest revising this paragraph:
> > > In some cases, the performance at n = 0 is better than when we include additional fine-tuning samples (n > 0). In these cases, the models may benefit from a di fferent method of fine-tuning which causes less forgetting than learning a new model on top of a pre-trained feature encoder however, in the interest of consistency across datasets we keep the fine-tuning procedure the same for all experiments. The trends noted above remain clear, though, whether directly applying the base model or training with additional samples from the test domain.
> >
> > The final statement reads that the authors are claiming that their approach would hold in the scenario where a different fine-tuning method is used, which there is no evidence for.  Please refine this last sentence to only claim that the trend holds under the chosen fine-tuning procedure (and that you hypothesize that it would hold in other cases if you wish).

---

> > > ### Author Response · Authors · 2024-07-15
> > >
> > > Thank you for your reply. We have revised the last sentence to read:
> > > > Since the baseline methods also experience a proportional drop in performance in these cases, the performance of our method under the fine-tuning procedure presented here still follows the same trends noted above. Other methods of fine-tuning should be explored in future work.

---

### Review · Reviewer_3F1X · 2024-06-05

**Summary Of Contributions:**

*Context:* I reviewed an earlier version of this work (https://openreview.net/forum?id=Vyw437epFz), which was rejected with the suggestion for a major revision. Having re-read the revised paper, the below review is focused on the delta to the previous version.

The main changes made for the revision appear to be relatively minor, and focused on changing individual wordings to remove causal terminology, e.g.,
- causal graph / causal model --> attribute graph / graphical model
- causal mechanism ---> probabilistic mechanism
- causal reasoning / information --> Bayesian reasoning / information
- causal relations --> statistical relations

While this is a step in the right direction, it arguably falls somewhat short of the requested "adjustment of the framing". Quoting from the previous AE decision:
> In some sense, instead of asking "how was this data generated?", the paper asks "what is this pre-trained generative model doing?".

This is a spot-on summary, but something that---in my opinion---is still is not sufficiently clear in the revised version.

I still get the impression that the learnt "attribute graph" is also meant/presented/interpreted as conveying something about true relations between attributes. However, one aspect that deserves more emphasis/attention is the fact that the learnt "attribute graph" also crucially depends on the pre-trained attribute estimators, and not just on the generative model (Enc & Dec), as the title might suggest. The attribute graph should thus be viewed as a summary of how changes to the *learnt* latent components translate to changes in the *predicted/estimated* attributes, assuming these changes are mostly mediated through influences among the attributes, rather than through a direct influence from latents on attributes.

If the main goal is to understand the interplay between the generative/latent variable model and the learnt attribute predictors, why not simply learn, say, a bipartite graph consisting only of arrows $z_i \rightarrow a_j$?

Another issue that deserves further discussion is the reliability of the fixed, learnt components under perturbations to individual latents in light of out-of-support issues: even though the perturbed $\tilde z_i$ are sampled such that they are marginally in-support, the resulting joint vector $\bf \tilde z$ may be out-of-support. As a result, both the Dec/Enc and the attribute predictors may be completely unreliable on the perturbed  $\bf \tilde z$ and $\bf \tilde x$.

All in all, it seems that many choices are somewhat arbitrary but presented in a way that could suggest that they follow in a principled way. However, the main evidence presented in favour of the proposed method is empirical. Thus, I would favour a framing along the lines of "this is just one way of approaching this problem, and there is no reason why this should work, or should be the best way to go about this problem, but it turns out that this method is actually effective and yields good results on real data on a relevant problem".

I list some further questions below:
_____

- In the perturbation procedure (Alg. 2, which should be moved to the main paper), why do you opt for (unregularized) OLS + post-hoc thresholding of the weights, instead of a sparse regression approach such as LASSO/L1 regularization?
- How is the standardization in Step 5 of Sec. 3.2.1 done? I could not find these details in Alg. 2. Why is standardization necessary / how does it impact the behaviour in combination with the subsequent thresholding?
- In Steps 3 & 4 of Sec. 3.2.1, why are the attributes estimated after first decoding and then re-encoding the perturbed latent? That is, why apply the attribute estimator to $Enc(\tilde x)=Enc(Dec(\tilde z)$ rather than to $\tilde z$ directly? If Enc inverts Dec, the two should be equivalent, but this is generally not the case even in-support, and certainly not out-of-support.
- In Stage 3 of Figure 2, why add both edges from $A_2,A_3$ to $A_4$. Wouldn't one be enough? If sparsity is desired, adding the single edge $z_2 \rightarrow A_4$ could also explain the dependence. Why not opt for this? (Related to my point about learning a bipartite graph)

**Audience:**

Yes

**Broader Impact Concerns:**

---

**Claims And Evidence:**

No

**Requested Changes:**

----

**Strengths And Weaknesses:**

---

---

> ### Author Response · Authors · 2024-06-21
>
> > However, one aspect that deserves more emphasis/attention is the fact that the learnt "attribute graph" also crucially depends on the pre-trained attribute estimators, and not just on the generative model (Enc & Dec), as the title might suggest.
>
> This is a valid point although the attribute estimators themselves are just simple models on top of the latent encodings of the generative model. It is also important to note that the vast majority of attributes in our experimental use case are descriptors which are exactly computed on the sequences.
>
> > The attribute graph should thus be viewed as a summary of how changes to the learnt latent components translate to changes in the predicted/estimated attributes, assuming these changes are mostly mediated through influences among the attributes, rather than through a direct influence from latents on attributes.
>
> This is correct and an excellent summary of our framework. Thank you for suggesting. We have added this exact sentence to the manuscript now to clarify this point.
>
> > If the main goal is to understand the interplay between the generative/latent variable model and the learnt attribute predictors, why not simply learn, say, a bipartite graph consisting only of arrows $z_i \rightarrow a_j$?
>
> As noted above, our goal is to understand more than the relationship between latents and attributes. We wish to learn the relationship between attributes themselves through the lens of the latents derived by the generative model.
>
> > Another issue that deserves further discussion is the reliability of the fixed, learnt components under perturbations to individual latents in light of out-of-support issues: even though the perturbed $\tilde z_i$ are sampled such that they are marginally in-support, the resulting joint vector $\bf \tilde z$ may be out-of-support. As a result, both the Dec/Enc and the attribute predictors may be completely unreliable on the perturbed $\bf \tilde z$ and $\bf \tilde x$.
>
> Each perturbation only changes one latent dimension. In other words, the $\Delta z$ is all zeros except one value. The perturbed value is sampled such that it is always within the bounds seen in training.
>
> Since the Enc/Dec is a VAE parameterized by a Gaussian distribution, any such perturbed z is in the support.
>
> > All in all, it seems that many choices are somewhat arbitrary but presented in a way that could suggest that they follow in a principled way. However, the main evidence presented in favour of the proposed method is empirical. Thus, I would favour a framing along the lines of "this is just one way of approaching this problem, and there is no reason why this should work, or should be the best way to go about this problem, but it turns out that this method is actually effective and yields good results on real data on a relevant problem".
>
> We have now added the following sentence in the conclusion to address this point: "We have provided here a domain generalization method that utilizes latent-mediated influences among molecular attributes and have empirically shown its efficacy to predict molecular attributes by learning from limited out-of-distribution data."
>
> > In the perturbation procedure (Alg. 2, which should be moved to the main paper), why do you opt for (unregularized) OLS + post-hoc thresholding of the weights, instead of a sparse regression approach such as LASSO/L1 regularization?
>
> Algorithm 2 has been moved to the main paper (now called Algorithm 1).
>
> We agree that there is more than one way to learn sparse weights and we wish to explore those in future work. One reason for not using LASSO on the full matrices of $\Delta Z$, $\Delta A$ was computational intractability due to the size of the matrices. On the other hand, for OLS we learn on a per-latent dimension basis independently (since only one latent dimension is perturbed at a time) but this is not possible if L1 regularization is added.
>
> > How is the standardization in Step 5 of Sec. 3.2.1 done? I could not find these details in Alg. 2. Why is standardization necessary / how does it impact the behaviour in combination with the subsequent thresholding?
>
> Standardization is done simply by centering the values to zero mean and scaling to unit variance. This has been added to the Algorithm block. This is done to normalize the response of each attribute since attributes may have vastly different ranges. This in turn leads to weights which can be compared against each other (and to a sparsity threshold).

---

> ### Author Response · Authors · 2024-06-21
>
> > In Steps 3 & 4 of Sec. 3.2.1, why are the attributes estimated after first decoding and then re-encoding the perturbed latent? That is, why apply the attribute estimator to $Enc(\tilde x)=Enc(Dec(\tilde z)$ rather than to $\tilde z$ directly? If Enc inverts Dec, the two should be equivalent, but this is generally not the case even in-support, and certainly not out-of-support.
>
> We estimate most attributes directly from the sequence $\tilde x$, while the rest rely on a predictor model trained on $z$. In general, the attribute space is discrete in this way — it only makes sense to discuss attributes of  sequences (or encodings thereof). Therefore, we use $\Delta z$ to sample nearby sequences and compare the attributes of those sequences by re-encoding them.
>
> > In Stage 3 of Figure 2, why add both edges from $A_2,A_3$ to $A_4$. Wouldn't one be enough? If sparsity is desired, adding the single edge $z_2 \rightarrow A_4$ could also explain the dependence. Why not opt for this? (Related to my point about learning a bipartite graph)
>
> Given weight matrix $W$, we know that $z_2$ influences $A_2$, $A_3$, and $A_4$. However, we also know that $A_4$ must be downstream of $A_2$ and $A_3$ since $z_3$ influences $A_4$ and not $A_2$ or $A_3$. A single arrow from $z_2 \rightarrow A_4$ would imply $z_2$ only influences $A_4$ (like $z_3$) since $A_2$ and $A_3$ are upstream of it, which is incorrect.
>
> Since we are unable to disambiguate the ancestral ordering of $A_2$ and $A_3$, we must include both edges to $A_4$. A single edge from $A_2 \rightarrow A_4$ or $A_3 \rightarrow A_4$ would lose information. Imagine the new Markov blanket of $A_4$. It would no longer include one of those two attributes since it would no longer be a direct parent of $A_4$.
>
> The transitive reduction operator will remove the edge from $A_1 \rightarrow A_4$, however, since we know $A_1$ must be an ancestor of $A_2$ and $A_3$.

---

> ### Comment · Reviewer_3F1X · 2024-06-27
> **Response to Authors' Response**
>
> Thank you for providing further clarification and adopting some of my suggested modifications.
>
> > the attribute estimators themselves are just simple models on top of the latent encodings of the generative model. It is also important to note that the vast majority of attributes in our experimental use case are descriptors which are exactly computed on the sequences.
>
> Please add this to the manuscript. The current exposition in 3.1 states that the attribute estimators take z=Enc(x) as input. If this is often not the case in practice, this passage should be modified accordingly. [If the attribute estimators take x as input, they can still apply Enc as a first step if required, but this would allow more flexibility].
>
> > One reason for not using LASSO on the full matrices of $\Delta Z$, $\Delta A$ was computational intractability due to the size of the matrices. On the other hand, for OLS we learn on a per-latent dimension basis independently (since only one latent dimension is perturbed at a time) but this is not possible if L1 regularization is added.
>
> Please add this explanation to the manuscript.
>
> > Standardization [...] is done to normalize the response of each attribute since attributes may have vastly different ranges. This in turn leads to weights which can be compared against each other (and to a sparsity threshold).
>
> Please add this to the manuscript.
>
> > We estimate most attributes directly from the sequence $\tilde x$, while the rest rely on a predictor model trained on $z$. In general, the attribute space is discrete in this way — it only makes sense to discuss attributes of sequences (or encodings thereof). Therefore, we use $\Delta z$ to sample nearby sequences and compare the attributes of those sequences by re-encoding them.
>
> Please add this to the manuscript (related to the first point above), and provide further explanation on the sequential nature of attributes (last sentence) as this was not sufficiently clear from the current presentation.
>
> > Given weight matrix $W$, we know that $z_2$ influences $A_2$, $A_3$, and $A_4$. However, we also know that $A_4$ must be downstream of $A_2$ and $A_3$ since $z_3$ influences $A_4$ and not $A_2$ or $A_3$. A single arrow from $z_2 \rightarrow A_4$ would imply $z_2$ only influences $A_4$ (like $z_3$) since $A_2$ and $A_3$ are upstream of it, which is incorrect.
>
> I was not suggesting a single arrow $z_2 \rightarrow A_4$, but rather $z_2 \rightarrow \{A_2, A_3, A_4\}$. (In principle, your assumptions do not exclude this option since each z can influence multiple attributes.)
>
> > Since we are unable to disambiguate the ancestral ordering of $A_2$ and $A_3$, we must include both edges to $A_4$. A single edge from $A_2 \rightarrow A_4$ or $A_3 \rightarrow A_4$ would lose information. Imagine the new Markov blanket of $A_4$. It would no longer include one of those two attributes since it would no longer be a direct parent of $A_4$.
> The transitive reduction operator will remove the edge from $A_1 \rightarrow A_4$, however, since we know $A_1$ must be an ancestor of $A_2$ and $A_3$.
>
> I understand the procedure. My point was rather that it relies on implicit assumptions that could be more clearly stated. E.g., something like "1. if an influence $z_i \to A_j$ can be explained through mediation via another $z_i\to A_k\to A_j$, prefer this option, i.e., add $A_k\to A_j$ and remove $z_i \to A_j$." and "2. if there are multiple potential mediators, add edges for each of them." I think adding more formal, mathematical desiderata & assumptions would further clarify the goal of finding maximally attribute-mediated relations. Perhaps adding some of the above discussion (e.g., to an appendix) could also be helpful to readers.

---

> > ### Author Response · Authors · 2024-07-15
> >
> > Thank you for your reply. We have added all of your latest suggestions to the revision.

---

### Decision · Action_Editor_Xg9U · 2024-07-29

**Recommendation:** Accept as is

**Comment:**

This is a resubmission of a previous TMLR submission (https://openreview.net/forum?id=Vyw437epFz). In the revised submission, the authors have carefully clarified their results and situated them within the context of causal machine learning. Although the proposed attribute graph is non-causal in nature, it is nevertheless useful for downstream tasks as illustrated in the experiments. After revision, all three reviewers are in favour of acceptance.

**Audience:**

Yes, see above.

**Claims And Evidence:**

The basic idea behind this paper is to learn a structural equation model that is implied by a pre-trained generative model. Instead of trying to replicate the causal reality behind the training data, the authors take the generative model as given, and try to explain what is going on inside the generative model. In some sense, instead of asking "how was this data generated?", the paper asks "what is this pre-trained generative model doing?". Thus, instead of estimating a causal model for the data, the authors are estimating a graphical model for a pre-trained model, and illustrate how this can be used in applications.